# From In Vitro Data to In Vivo Interspecies Danger Communication: A Study of Chemosensing via the Mouse Grueneberg Ganglion

**DOI:** 10.3390/ani12030356

**Published:** 2022-02-01

**Authors:** Ana Catarina Lopes, Julien Brechbühl, Flavio Ferreira, Marjorie Amez-Droz, Marie-Christine Broillet

**Affiliations:** Department of Biomedical Sciences, Faculty of Biology and Medicine, University of Lausanne, Bugnon 27, CH-1011 Lausanne, Switzerland; anacatarina.lopes@unil.ch (A.C.L.); julien.brechbuhl@unil.ch (J.B.); flavio.ferreira@unil.ch (F.F.); marjorie.amez-droz@unil.ch (M.A.-D.)

**Keywords:** olfaction, Grueneberg ganglion, kairomones, TAS2Rs, predators, danger detection, chemical communication

## Abstract

**Simple Summary:**

The mouse olfactory system is essential for danger detection with a critical role in the Grueneberg ganglion subsystem. This organ, which is localized at the tip of the nose, is implicated in the recognition of kairomones, or chemical cues released by predators which allow interspecies communication. These kairomones, which are present in the secretions of predators, will induce fear-related behaviours in mice. It is not yet known how the Grueneberg ganglion neurons can detect these molecules; however, three specific bitter taste receptors, known as TAS2Rs, that are present in the Grueneberg ganglion play a role in this detection. Here, using in vitro, ex vivo and in vivo experimental approaches, we identified two novel and potent sources of kairomones that are recognized by the mouse Grueneberg ganglion neurons, namely the biological secretions from the raccoon (*Procyon lotor*) and the skunk (*Mephitis mephitis*).

**Abstract:**

In the wild, mice have developed survival strategies to detect volatile cues that warn them of potential danger. Specific olfactory neurons found in the Grueneberg ganglion olfactory subsystem can detect alarm pheromones emitted by stressed conspecifics, as well as kairomones involuntarily released by their predators. These volatile chemical cues allow intra- and interspecies communication of danger, respectively. Alarm pheromones, kairomones and bitter taste ligands share a common chemical motif containing sulfur or nitrogen. Interestingly, three specific bitter taste receptors (TAS2Rs) have been found in the Grueneberg ganglion neurons that are implicated in danger signalling pathways. We have recently developed a TAS2R–expressing heterologous system that mimics the Grueneberg ganglion neuron responses after kairomone stimulation. Here, we demonstrated by in vitro, ex vivo and in vivo experiments that the biological secretions from the raccoon (*Procyon lotor*) and the skunk (*Mephitis mephitis*) were acting as potent sources of kairomones. They activated the Grueneberg ganglion neurons and induced fear-related behaviours in mice. Identification of new sources of semiochemicals is a first step towards an understanding of the interspecies danger communication that takes place in the Grueneberg ganglion.

## 1. Introduction

In the wild, animals need to communicate and exchange information related to the environment for the benefit of their species’ survival. One way of communicating is through chemosensation, which allows for the detection of chemicals present in the environment. Molecules carrying a specific chemical message are called “semiochemicals”.

Among these “semiochemicals” are pheromones, volatile and non-volatile active substances present in urine, sweat, exocrine glands and/or in mucus secretions of genitals, that will induce different physiological and behavioural responses in individuals of the same species [1]. Intraspecies chemical communication via pheromones plays important roles in social and reproductive behaviours [2].

Alarm pheromones are an important class of pheromones. They are essential for detecting and sensing danger as well as inducing a repulsive behaviour [3]. Indeed, to survive and protect themselves from a danger such as the presence of predators, mammalian prey, such as mice, have developed abilities to sense the alarm pheromones emitted by stressed or injured conspecifics. These alarm pheromones will induce defensive and vigilant behaviours in the conspecific receiver [4].

The presence of a danger can also be transmitted by another group of volatile chemical substances called kairomones. Kairomones are defined as chemical substances carrying a message that will be detected by another species [5]. Rodents, such as mice, can sense involuntarily released kairomones in the environment by the predators themselves, allowing for interspecies communication [6]. Kairomones can be present, similar to alarm pheromones, in the urines, anal glands or faeces of the predators [7]. These predator cues will be beneficial for the survival of the receivers as they alert the receivers to dangerous situations.

The olfactory system detects these danger warning cues [8]. It is composed of different subsystems: the main olfactory epithelium, the vomeronasal organ, the Grueneberg ganglion (GG) and the septal organ [9]. These olfactory subsystems have different functional specializations [10,11]. Neurons from the GG, for example, specifically detect alarm pheromones and kairomones, which induce fear-related behaviours in mice [12]. Interestingly, the chemical substances recognized by GG neurons share a specific chemical structure [13]. More precisely, the identified mouse alarm pheromone, 2-*sec*-Butyl-4,5-dihydrothiazole (SBT), showed similarities with predator scents (such as 2,5-dihydro-2,4,5-trimethylthiazoline (TMT), which is found in fox (*Vulpes vulpes*) faeces) [13]. In fact, GG neurons are activated by chemical substances belonging to pyrazine/pyridine derivatives, which are a specific class of volatile heterocyclic sulfur- or nitrogen-containing compounds that induce avoidance and fear-related behaviours [14,15].

Remarkably, the conserved chemical structure recognized by GG neurons is related to a class of bitter ligands, such as 2-etylpyrazine (2-EP). These bitter ligands warn the mice against ingestion of the toxic substances present in food and act on an identified and specific family of G-protein-coupled receptors: the family of bitter taste receptor 2 (TAS2Rs) [16]. These receptors are not only present in the tongue but also in tissues from different internal organs [17]. Moreover, three specific TAS2Rs and their G-protein α-gustducin (GNAT3) have been found in the mouse GG, namely the TAS2R115, TAS2R131 and TAS2R143 receptors [18].

Studies in vitro can be used to identify the potential ligands associated with membrane receptors. The receptors and the G-protein subunit α-gustducin can be co-expressed at the membrane of human embryonic kidney cells 293 (HEK 293) [19]. Different ligand candidates can then be tested by observing the intracellular calcium variations in the cells upon exposure. Thanks to the calcium imaging technique, it has been shown that TAS2R143 showed activity in the presence of the mouse alarm pheromone SBT, the bitter ligand 2-EP, the kairomone TMT and the kairomone 2-propylthietane (2-PT) from the stoat (*Mustela erminea*), which all share the same structure as SBT and TMT [18]. The spectrum of ligands that can activate the two other GG receptors (TAS2R115, TAS2R131) has yet to be defined. To try to identify potential ligands for these receptors, predator secretions can be directly tested in vitro on TAS2Rs-expressing HEK cells and ex vivo on mouse GG neurons. Their effects can be compared to those observed in the presence of the mountain lion (*Puma concolor*) urine, which activates GG neurons and induces fear-related behaviours in mice as we have previously demonstrated [15].

Our objective, then, was to identify new sources of kairomones released by predators using in vitro, ex vivo and in vivo approaches. Two distinct animal secretions were selected: raccoon (*Procyon lotor*) urine and secretions from the anal scent glands of skunks (*Mephitis mephitis*). Raccoons are predators that kill and eat mice [20]. Analysis of their urine has identified different types of molecules, such as volatile hydrocarbons, sulfur and nitrogen compounds [21]. The secretions from the anal scent glands of the skunk were selected due to the specific sulfuric odour emitted by their spray, which is used by the skunk as a defence mechanism [22]. The skunk is also considered a mouse predator. Here, we found that both predator secretions act as potent sources of kairomones, which are recognized by GG neurons and induce fear reactions in mice.

## 2. Materials and Methods

### 2.1. In Vitro Experiments

#### 2.1.1. Cell Culture

Human embryonic kidney (HEK) 293 cells were cultured in Dulbecco’s Modified Eagle Medium (Gibco^TM^ DMEM; Thermo Fisher Scientific, Waltham, MA, USA), supplemented with 0.2% gentamycin antibiotic (50 mg/mL; Gibco^TM^; Thermo Fisher Scientific, MA, USA) and 10% of foetal bovine serum (Gibco^TM^ FBS; Thermo Fisher Scientific, MA, USA). Cells were cultured in T-25 flasks (Falcon^®^; VWR, Radnor, PA, USA) and incubated in a 5% CO_2_ atmosphere at 37 °C.

#### 2.1.2. HEK 293 Cell Transfection

The plasmids used for the experiments were constructed according to Moine et al. [18]. In short, the bitter taste receptors TAS2R115, TAS2R131, TAS2R143, as well as a gustducin G protein α-subunit, were respectively cloned in a mammalian expression vector (pcDNA3.1; GenScript) containing a tag in its N-terminal position with amino acids from the bovine rhodopsin receptor, allowing for their expression at the cell membrane. The gustducin G protein α is a chimera of the human Gα16 with 44 amino acids from the mouse gustducin G protein α-subunit in its C-terminal [18]. Plasmid DNA was amplified and extracted from *Escherichia coli* bacteria. HEK 293 cells were cultured in 60 mm petri dishes (Falcon^®^; VWR, PA, USA) on a sterile microscope glass cover slip (22 × 40 mm; Menzel™; Thermo Fisher Scientific, MA, USA) for transfection. Cell media was changed and replaced with a new one 2–3 h before transfection. Cells were transfected via the calcium phosphate transfection method when the confluence reached 50–70% [23]. Cells were co-transfected with the Gα16 plasmid, the three TAS2Rs plasmids and with the GFP reporter plasmid to verify the efficiency of the transfection. Cells were used for calcium imaging 24–48 h after transfection.

#### 2.1.3. Calcium Imaging Experiments on Transfected HEK 293 Cells

The cover slip with HEK 293 cells was transferred into the calcium imaging chamber (RC-26; Warner Instruments, Holliston, MA, USA). Cells were loaded with Fura-2 acetoxymethyl ester (AM) dye (10 µM; Thermo Fisher Scientific) and pluronic acid (0.1%; Invitrogen^TM^ F-127; Thermo Fisher Scientific, MA, USA) mixed in Ringer solution (composed of 140 mM NaCl, 5 mM KCl, 10 mM HEPES, 2 mM CaCl_2_ and 1 mM MgCl_2_). Cells were incubated at 37°C in the dark for 30–45 min. Cells were continuously perfused with Ringer solution and observed with an inverted fluorescence microscope (ZEISS Axiovert 135) connected to a camera (Photometrics Scientific^®^ CoolSNAP ES2 camera; Visitron Systems GmbH, Puchheim, Germany), which allowed for the computational visualization via VisiView^®^ (Visitron Systems GmbH, Germany) software. Observations were performed at 480 nm (GFP) and 380 nm (Fura-2). Adenosine 5′-triphosphate in Ringer solution (100 µM; Sigma-Aldrich^®^ ATP; Merck, Schaffhausen, Switzerland) was perfused via a syringe at the beginning and end of the experiments to verify the viability of the cells. Biological secretions (PredatorPee^®^, Hermon, ME, USA and Pete Rickard Co., Galeton, PA, USA) collected from multiple individuals were mixed in Ringer solution [1/100; rabbit (*Oryctolagus cuniculus*), mountain lion (*Puma concolor*) and raccoon (*Procyon lotor*) urines] and injected via a syringe.

The secretions from the anal glands of the skunk (*Mephitis mephitis*) were first diluted in DMSO [ratio 1:2–3] due to their viscosity and then mixed in Ringer solution (1/100). All substances were tested on non-transfected and TAS2Rs-transfected HEK cells. As previously shown by Moine et al. [18], the transfected cells were sensitised to predator secretions (the mountain lion urine, Fisher’s exact test ***; the raccoon urine, Fisher’s exact test *** and the secretions from the skunk, Fisher’s exact test **) and not to non-predator secretions (the rabbit urine, Fisher’s exact test ns). The percentage of responding HEK cells was obtained taking into account only the cells that responded to the viability tests (ATP perfusions). Responses to the biological secretions tested that corresponded to at least 10% of those generated by ATP (baseline activity) were included [24].

### 2.2. Ex Vivo Experiments

#### 2.2.1. Animals

OMP-GFP mice [25] pups (postnatal days P3–P8) were killed by decapitation in accordance with Swiss legislation and approved by the EXPANIM committee of the Lemanique Animal Facility Network and the veterinary authority of Canton de Vaud (SCAV).

#### 2.2.2. GG Tissue Slice Preparation

As described in Brechbühl et al. [13], pup heads were placed, for GG olfactory subsystem dissection, in ice-cold artificial cerebrospinal fluid (ACSF: 118 mM NaCl, 25 mM NaHCO_3_, 10 mM D-glucose, 2 mM KCl, 2 mM MgCl_2_, 1.2 mM NaH_2_PO_4_, 2 mM CaCl_2_ (pH 7.4) and gassed with oxycarbon (95% O_2_, 5% CO_2_). Tips of the noses were incorporated in 4% agar prepared in phosphate-buffered saline (PBS). GG tissue slices of 80 µm were then prepared on ice with a vibroslicer (VT1200S; Leica Biosystems, Welzlar, Germany). The slices were conserved in ACSF solution and selected under a fluorescent stereomicroscope (M165 FC, Leica) to visualize GG neurons, thanks to their GFP expression. As for the experiments in vitro, the GG tissue slices were loaded with Fura-2AM [10 µM] and pluronic acid [0.1%] mixed in ACSF. The slices were incubated for 45–60 min (37 ˚C, 5% CO_2_) and then placed in the calcium imaging chamber [13].

#### 2.2.3. Calcium Imaging Experiments

Calcium variations were observed with the same parameters as those for the HEK cells. The GG neurons were stimulated with pure animal secretions diluted in the ACSF solution [1/100]. As for the HEK cells, the secretions of the anal glands of the skunk were first diluted in DMSO [ratio 1:2–3] and then mixed in ACSF solution [1/100]. To test the viability of the neurons, ACSF-KCl (35 mM NaCl, 25 mM NaHCO_3_, 10 mM D-glucose, 100 mM KCl, 3 mM MgCl_2_, 1.2 mM NaH_2_PO_4_, 0.5 mM CaCl_2_ gassed with oxycarbon 95% O_2_ and 5% CO_2_) was perfused at the beginning and end of the experiment. Neurons that responded to the ACSF-KCl and generated responses from the biological secretions that corresponded to at least 10% of the responses to ACSF-KCl (baseline activity) were selected to calculate the percentage of responding neurons [13,24].

### 2.3. In Vivo Experiments

#### 2.3.1. Animals

Six male adult C57BL/6 mice aged between 14 and 19 months (*Mus musculus*; Janvier Labs) were used for behavioural tests in accordance with Swiss legislation and approved by the EXPANIM committee of the Lemanique Animal Facility Network and the veterinary authority of Canton de Vaud (SCAV).

#### 2.3.2. Behavioural Test

Experimental procedures were adapted from Brechbühl et al. [15]. An open field test was used to detect and observe fear-related behaviours of six mice in the presence of the biological secretions. Mice were housed in a room with a 12:12 h light/dark inverted cycle in their home cage with food and water ad libitum. One week before the beginning of the experiments, the animals were familiarized with the test arena to minimize environmental stress. The test arena consisted of a closed Plexiglas box (45 × 25 × 19 cm) and a piece of blotting paper (3 × 3 cm) in one of the corners. Blotting paper of the size of the box covered the floor to avoid direct contact of the mice with the Plexiglas. Mice behaviours were recorded for at least 5 min during the nocturnal phase from the top of the arena covered with a Plexiglas plate by IR-sensitive HD camera under infrared vision. Video recordings were analysed with ANY-maze software (Stoelting Europe, Dublin, Ireland), a video tracking system detecting the centre of the animal as reference point, and the following parameters were quantified: the total walking distance, the number of entries in the central or danger zone, as well as the total time of immobility (immobility sensitivity of 65% with a minimum immobility period threshold of 1000 ms). Each mouse was presented with one stimulus/cue per day [15]. The non-fear inducing cues (water–rabbit urine) were presented randomly among the fear-inducing cues (mountain lion, raccoon and skunk secretions). Test sessions were performed at least two hours after control sessions.

### 2.4. Statistical Analyses

Excel 2019 and GraphPad 9 (Prism^®^, San Diego, CA, USA) were used to generate bar graphs and perform the statistical analyses. Statistical analyses were performed using a two-sided Fisher’s exact test for the in vitro experiments comparing non-transfected cells versus TAS2Rs-expressing cells for each cue. Comparisons between non-predator versus predators were performed using an unpaired two-tailed Student’s *t*-test or a Mann–Whitney test, in the case of non-respect of the homoscedasticity (Fisher’s F-tests). A one-tailed paired Student’s *t*-test was used to compare the control sessions (water) and the session test (conditions) for the in vivo experiment. Values are expressed as mean ± SEM with scatter dot plots. Significance levels are indicated as follows: * *p* < 0.05; ** *p* < 0.01; *** *p* < 0.001; ns, not significant.

## 3. Results

### 3.1. TAS2Rs-Expressing HEK Cells Respond to Predator Secretions

We first tried to identify potent natural sources of mouse kairomones by testing two different biological secretions from mice predators in an in vitro model, thus, minimizing the number of mice used according to the “3Rs” principle (replacement, reduction and refinement) [26]. It is also less expensive and less time-consuming as a basic screening to verify that the substances that will be used further are potential candidates before working directly with mice.

We thus used an established heterologous system for the expression of the Grueneberg ganglion TAS2Rs receptors [18], in which we co-transfected the three TAS2Rs (TAS2R115, TAS2R131 and TAS2R143 or each receptor individually) with the chimera gustducin G protein α. A GFP reporter plasmid was also used to control the efficacy of cellular transfection. Calcium imaging experiments were then performed on cells to observe the variations in the intracellular calcium after perfusions of the biological secretions (Figure 1).

We selected biological secretions from four different animals. The mountain lion (*Puma concolor*) urine was selected as our positive control because we knew already from our previous experiments that it could activate GG neurons and induce fear-related behaviours in mice [15]. The biological secretions of interest were coming from the raccoon (*Procyon lotor*) and the skunk (*Mephitis mephitis*). To compare predator versus non-predator secretions, we chose the urine of rabbit (*Oryctolagus cuniculus*), an herbivorous animal.

The urines from the rabbit, the mountain lion and the racoon, as well as the secretions from the anal glands of the skunk, were perfused on non-transfected HEK (Figure 1a) and on TAS2Rs-transfected HEK cells (Figure 1b) at a dilution of 1/100 in Ringer solution. Interestingly, we observed that transfected HEK cells with TAS2Rs showed a significant increase in their intracellular calcium concentration (Figure 1c) in the presence of the mountain lion urine and racoon urine, as well as in the presence of the anal gland secretions of the skunk. On a peculiar note, the intracellular calcium increases in response to the mountain lion and raccoon urines observed differ from the response of the secretions from the skunk. Indeed, the response of the latter appeared to last longer but was still reversible (Figure 1b).

TAS2R115 and TAS2R131 are orphan receptors, thus, we decided to express each receptor individually in our heterologous system to verify the presence of the calcium increases induced by the different biological secretions (Figure 1d). Interestingly, the sensitivity range of the three TAS2R receptors was different. Namely, the TAS2R115-expressing cells showed a higher sensitivity towards skunk secretions; whereas TAS2R131 responded preferentially to the raccoon urine. Alternatively, TAS2R143 did not show any significative difference once exposed to non-predator versus predator secretions, thereby indicating that the kairomones present in these secretions do not activate this particular TAS2R (Figure 1d). Taken together, these results showed a differential contribution of the three TAS2Rs in kairomone detection.

With these in vitro results, we have demonstrated that the biological secretions from the raccoon and the skunk, two animals which are mice predators, can activate TAS2R-expressing cells [27,28].

### 3.2. Mouse GG Neurons Respond to Raccoon Urine and Skunk Anal Secretions

The next step was to verify if the raccoon urine and the anal secretions from the skunk could activate mouse GG neurons in an ex vivo preparation. Calcium imaging experiments were then performed on GG neurons using tissue slices from the GG of OMP-GFP mice, where the reporter GFP is under the control of the olfactory marker protein (OMP) promoter, which allows for the visualization of all mature olfactory sensory neurons in green [25]. Mature olfactory sensory neurons are known to be activated by kairomones [29].

As expected, the mountain lion urine induced a significant intracellular calcium increase (Figure 2a). An intracellular calcium increase was also observed when we perfused the raccoon urine and the secretions of the anal glands of the skunk, as we can see in the representative calcium transients. Moreover, we observed, as with the in vitro results, long-lasting responses to the secretions from the skunk compared to the two other body fluids.

The percentage of responding GG neurons was calculated for each condition with several independent experiments. Ex vivo results with the rabbit urine were similar to those of our in vitro experiments. Indeed, we observed only a small proportion of GG neurons responding (4.7%) to the urine. For the predator secretions, we observed a significant increase in the percentage of responding GG neurons. Indeed, nearly all GG neurons were activated by the mountain lion urine (95.9%). Compared to the results in vitro, GG neurons seem to be more sensitive to the skunk secretions than to the raccoon urine (79.9% versus 49.1% of responding GG neurons) (Figure 2b). This can potentially be explained by the multiple signalling pathways that take place in true GG neurons [30].

In summary, our ex vivo results correlate with our previously obtained in vitro results, indicating that the biological secretions from the raccoon and the skunk contain chemical molecules recognized by the neurons of the mouse Grueneberg ganglion.

### 3.3. Raccoon Urine and Skunk Secretions Induce Fear-Related Behaviours in Mice

Danger detection via the Grueneberg ganglion neurons induces fear-related behaviours in mice [31]. We therefore verified if raccoon and skunk biological fluids contain chemical molecules associated with danger cues, which are able to induce these fear-related behaviours in mice.

In an open field arena, we observed how mice reacted when exposed to: first, water, which is odourless and used as a control, and then, successively to the biological secretions for the first time. Mice could be tracked in the open field arena, and the total distances they walked could be traced. Each test condition was compared to its control (water) and quantified as an index for different parameters, such as the walking distance, the number of entries into the central zone and the danger zone as well as the immobility time.

The water and rabbit urine showed a homogenic exploration distance travelled by the mice around the arena, consistent with the fact that the rabbit is not a mouse predator. For the three predator secretions, a less homogenic exploration was observed. Indeed, the mice preferred to stay along the edges of the arena, away from the lower right corner, where the blotting paper with the biological secretions was present, as we can observe in the representative tracking distance of one mouse (Figure 3a).

We observed no significant differences between the water and the rabbit urine (−6 ± 19% and −12 ± 12%, respectively) when we looked at the walking distance index. No significant differences were also observed for the central entries index (−6 ± 29% and −12 ± 14%, respectively) or for the danger zone entries index (−21 ± 29% and +6 ± 28%, respectively). Interestingly, the mice seemed to be sensitive to the new smell of the rabbit urine, as we observed an increase in this parameter. The mice walked all around the arena; therefore, the immobility time index was not significatively different for water and rabbit urine (+2 ± 9 and +6 ± 7%, respectively).

Conversely, a significant decrease in the walking distance index was observed for the mountain lion urine, the raccoon urine and for the secretions from the skunk (−24 ± 9%, −25 ± 10% and −28.6 ± 8.3%, respectively). In addition to walking less, the mice were also less present in the central zone, indicating that they were anxious in the presence of predator secretions. The urine of the mountain lion and the secretions of the anal glands of the skunk showed a significative decrease in presence in the central zone (−25 ± 8% and −33 ± 12%, respectively). Surprisingly, the raccoon urine showed a non-significant decrease (−23 ± 12% versus −6 ± 29% observed for the water for example) but a *p*-value of 0.08 close to the significance. This is due to the heterogeneity of the distribution, which is more homogenic for the mountain lion and the skunk. Moreover, the mice prefer to stay away from the danger zone as secretions from the mountain lion, the raccoon and the skunk induced significative avoidance of the danger zone (−62 ± 18%, −48 ± 19% and −71 ± 17%, respectively).

In the presence of the three predator urines, the mice walked a shorter distance, were less present in the central or danger zone and had also a higher immobility time index. Indeed, for this last parameter, we observed a significative decrease for the mountain lion urine, the raccoon urine and the secretions of the anal glands of the skunk (+16 ± 5%, +28 ± 8% and +22 ± 6% respectively) (Figure 3b).

In summary, these results obtained in vivo are correlated with the previous ones obtained in vitro and ex vivo. They confirm that the raccoon urine and the secretions of the anal glands of the skunk induce fear-related behaviours in mice, and, therefore, must contain kairomones recognized by the neurons of the mouse Grueneberg ganglion.

## 4. Discussion

Chemical communication between species is crucial for survival. The Grueneberg ganglion (GG), an olfactory subsystem, is specialized in mice in intra- and interspecies communication because it can detect alarm pheromones and kairomones, respectively, and warn the mice about a danger [11,32], such as the presence of predators. Here, we studied the effect of two novel biological secretions as potent sources of kairomones using an in vitro, ex vivo and in vivo integrative GG-based strategy.

First, thanks to a heterologous HEK cell system that expressed the TAS2Rs, we were able to investigate different biological secretions, which allowed us to screen their kairomonal effects. Here, our biological sources of interest were the raccoon urine (*Procyon lotor*) and the secretions of the anal glands of the skunk (*Mephitis mephitis*). These two animals are mice predators. Testing these biological secretions first in an in vitro model, we observed that these selected source of kairomones can activate our heterologous system, mimicking GG neurons. Interestingly, it appears that the three TAS2Rs display, in a heterologous system, a differential range of detection that might be important for improving their chemosensory complementarity.

How these TAS2Rs contribute to the general kairomone detection in the wild remains to be demonstrated. Indeed, the bitter taste signalling cascade is not the only pathway present in GG neurons. Transmembrane guanylyl cyclase (GC) subtype GC-G pathway could also play a role in the detection of chemical danger cues since it is already known that it can play a role as an alarm pheromone receptor [33,34]. We can assume that, while the molecules present in the raccoon’s urine are mostly recognized by TAS2Rs, some molecules present in the secretions from the skunk can have a similar structure with alarm pheromones and also be recognized by the GC-G pathway. This could be an explanation for the differences observed in vitro and ex vivo; however, the two biological sources tested must have several kairomones in common recognized by TAS2Rs.

Interestingly, the long-lasting responses with the skunk secretions are the same in the two models. This kind of response can also be observed in primarily in other studies testing sulfuric molecules and their impact (for example, in main olfactory epithelium (MOE) subsystem) [35] Knowing that skunk spray is a strong defensive protection for them in the wild, this is unsurprising. The defensive spray produced by their glands contains specifical thiols volatile molecules that can last over time [36]. This long-lasting emission is reflected therefore in our in vitro and ex vivo responses after the injection of skunk secretions.

Kairomones released unintentionally by the predators induce fear-related behaviours in mice, which allows them to avoid danger. Therefore, to be a novel source of kairomones, these biological secretions should induce these behaviours in an in vivo model. Our results demonstrated that the raccoon urine and the secretions from the skunk induced these fear-related behaviours, which were not observed for the non-predator rabbit urine. This implies that these two biological sources contain kairomones, not yet chemically identified, which are recognized by the Grueneberg ganglion neurons. In correlation with our ex vivo results, we observed that the secretions from the skunk appeared to be strongly aversive for all the mice, which was less so the case for the raccoon urine, where we observed more heterogeneity between the mice; however, the two biological secretions both showed significative results.

In this study, we focused on the Grueneberg ganglion olfactory subsystem; however, the same in vitro, ex vivo and in vivo studies can be applied to other olfactory subsystems, such as the MOE. Trace amine-associated receptors (TAARs) are expressed in the GG and also in the MOE. Ligands recognized by these receptors are small volatile amine-derived molecules that can be found in natural secretions and that mediate innate odour aversion or attraction behaviours [37].

Body fluids are composed of a mix of molecules [6]. It would be interesting to perform a complete chromatography analysis to determine their chemical composition. Once the precise chemicals are identified, it will be possible to repeat the different experiments performed in our study with a special focus on the selectivity of the olfactory system involved using axotomized mice, for example [32,34].

In addition, we will be able to verify whether there are common chemical molecules in the raccoon urine, the skunk secretions or the mountain lion (*Puma concolor*) urine [15]. Indeed, as the predator secretions induced similar activations of TAS2Rs-expressing HEK cells, activations of the Grueneberg ganglion neurons, as measured by intracellular calcium, increase, and, more importantly, they induced similar fear-related behaviours. Finding a common volatile molecule recognized specifically by the three TAS2Rs present in the Grueneberg ganglion neurons will strengthen understanding of their role in danger detection and provide a better understanding of the predator–prey chemical communication.

From an environmental point of view, the identification of potent sources of kairomones will be helpful for the development of biological pest control molecules. Indeed, rodents, such as mice, are considered vectors of diseases that are potentially dangerous for human health. They are also a source of economic loss via food consumption and contamination [38]. Specific kairomones, such as TMT can be used to reduce the number of offspring born, rather than increasing the mortality rate induced by rodenticides [39].

Kairomones, such as TMT, are often used as a stressing stimulus for rodents in neuroscience studies for post-traumatic stress disorders or anxiety disorders [40]. Exposure to predator urines, or to the isolated kairomones from these urines, might produce the stress-induced behavioural, molecular and physiological alterations observed in these disorders [41]. Identification of new natural sources of kairomones will then be helpful to generate anxiety and test, for example, animal models that exhibit symptoms of neurologic disorders.

## 5. Conclusions

In this study, we have identified two potent and natural sources of kairomones: the raccoon urine and the secretions of the anal glands from the skunk. We demonstrated with in vitro experiments mimicking the Grueneberg ganglion neurons with their TAS2Rs expression, with ex vivo experiments on neuronal tissue slices and with in vivo behavioural experiments that these two biological fluids contain molecules recognized by the mouse Grueneberg ganglion. Furthermore, these two predator sources induce fear-related behaviours on mice, confirming the presence of molecules of danger that act as kairomones. Further research will need to be conducted in order to identify the precise chemical molecules present in these natural sources and to potentially determine specific ligands for each TAS2R. It would be interesting to find the common molecule(s) present in these secretions that could by themselves activate the Grueneberg ganglion neurons. Research on this olfactory subsystem is still open, and we need to understand the role and the mechanisms of chemical danger recognition in more detail, especially the function of the three TAS2Rs present in the neurons, which is essential for chemical detection and species survival.

## Figures and Tables

**Figure 1 animals-12-00356-f001:**
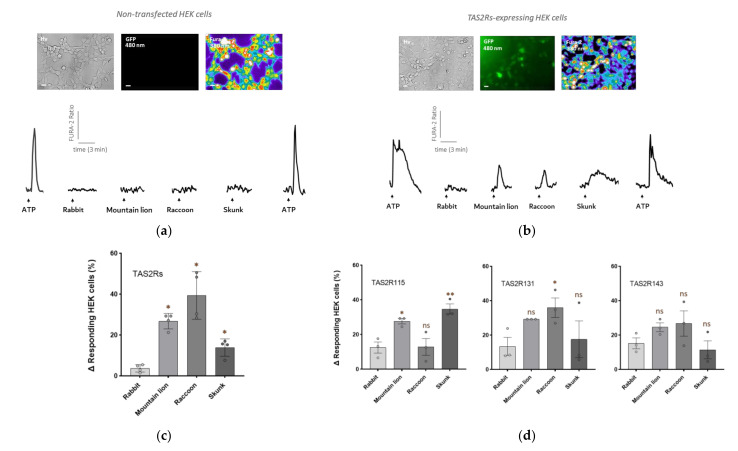
The urine of the racoon and the secretions from the anal glands of the skunk activate TAS2Rs-expressing HEK cells in an *in vitro* model. (**a**,**b**) Microscopic images of non-transfected HEK cells (**a**) and of TAS2Rs-transfected HEK cells (**b**) observed with a DIC contrast (Hv) at 480 nm (GFP) and at 380 nm after Fura-2 AM loading (Fura-2). Scale bars: 50 µm. Representative examples of the intracellular calcium changes observed in one non-transfected cell (**a**) and in one transfected cell (**b**) in the presence of ATP (perfused to control the viability of the cells) or in the presence of the urines of the rabbit (non-predator), of the mountain lion (predator), of the raccoon (predator) and of the secretions from the anal glands of the skunk (predator). The Fura-2AM ratio is given in arbitrary units and time in minutes. Arrows indicate the perfusion times. (**c**) Graph representing the delta percentage of responding HEK cells obtained by subtracting the number of non-transfected cells (control) from the number of responses of the TAS2Rs-expressing cells. *N* = number of experiments for each cue, *n* = number of cells in total: control (*N* = 4; *n* = 181); TAS2Rs (*N* = 4; *n* = 184). Values are expressed as mean ± SEM; unpaired Student’s *t*-test or Mann–Whitney test used to compare the non-predator versus predators, * *p* < 0.05 (in brown). (**d**) Graph representing the delta percentage of responding HEK cells obtained by subtracting the number of non-transfected cells (control) to the number of responses of the TAS2Rs-expressing cells. *N* = number of experiments for each cue, *n* = number of cells in total: control (*N* = 4; *n* = 181); TAS2R115 (*N* = 3; *n* = 105); TAS2R131 (*N* = 3, *n* = 123); TAS2R143 (*N* = 3; *n* = 119). Values are expressed as mean ± SEM; unpaired Student’s *t*-test or Mann–Whitney test used to compare the non-predator versus predators, * *p* < 0.05; ** *p* < 0.01; ns, not significant (in brown).

**Figure 2 animals-12-00356-f002:**
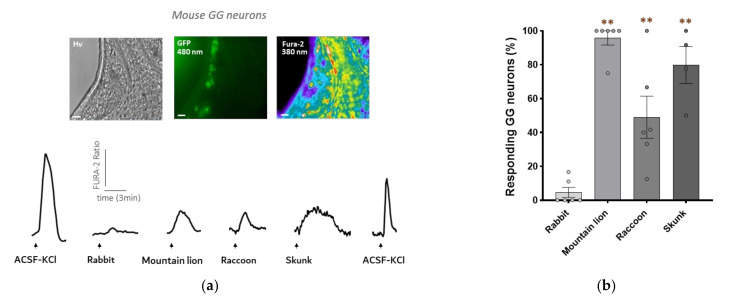
The urine of the racoon and the secretions from the anal glands of the skunk activate mouse GG neurons in an *ex vivo* tissue preparation. (**a**) Microscopic images of a tissue slice of a mouse GG observed with a DIC contrast (Hv) at 480 nm (GFP) and at 380 nm after Fura-2 loading (Fura-2). Scale bars: 50 µm. Representative examples of the intracellular calcium changes observed in one GG neuron in the presence of an ACSF-KCl solution perfused to control the neuronal viability or in the presence of the urines of the rabbit (non-predator), of the mountain lion (predator), of the raccoon (predator) and of the secretions from the anal glands of the skunk (predator). The Fura-2AM ratio is given in arbitrary units and time in minutes. Arrows indicating the perfusions time. (**b**) Graph representing the percentage of responding GG neurons to the different chemical cues. *N* = number of experiments for each condition; *n* = number of neurons in total for each condition: rabbit (*N* = 6; *n* = 38); mountain lion (*N* = 6; *n* = 49); raccoon (*N* = 6; *n* = 50); skunk (*N* = 4; *n* = 39). Values are expressed as mean ± SEM; unpaired Student’s *t*-test or Mann–Whitney test was used to compare the cues (non-predator versus predators), ** *p* < 0.01; ns, not significant (in brown).

**Figure 3 animals-12-00356-f003:**
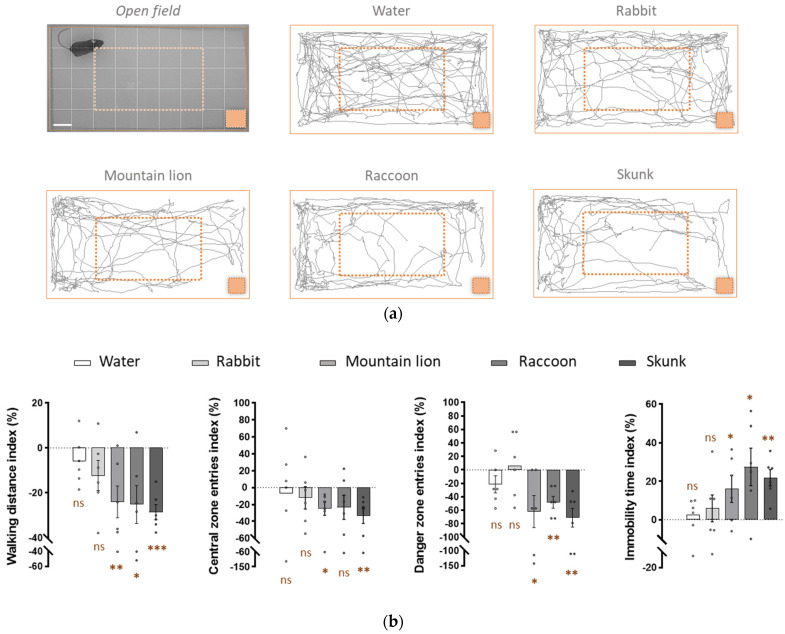
The urine of the racoon and the secretions from the anal glands of the skunk release potent fear-inducing kairomones for mice. (**a**) Behavioural open field arena (*Open field*) where water or biological secretions were present. The limits of the arena (full line rectangle in orange), the central zone (dashed line rectangle in orange) and the danger zone (blotting paper represented in orange, at the bottom right) are shown. Mice were tracked for 5 min, and 200 µL of the tested biological secretions were added on the blotting paper. Representative tracking distances are shown for the different conditions (water or biological secretions from rabbit, mountain lion, raccoon or skunk) for one mouse. Scale bar: 5 cm. (**b**) Quantification of the stress-related behaviours represented according to the control and displayed as indexes (%). The total walking distance, the number of visits in the central zone or in the danger zone and the immobility time were quantified. Significant innate fear reactions were observed for the mountain lion urine, the raccoon urine and the secretions of the anal glands of the skunk. Six adult mice were used. Values are expressed as mean ± SEM; one-tailed unilateral paired Student’s *t*-test was used, * *p* < 0.05; ** *p* < 0.01; *** *p* < 0.001; ns, not significant (in brown).

## Data Availability

The data presented in this study are available upon request to the corresponding author.

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
