# Peer review of "From In Vitro Data to In Vivo Interspecies Danger Communication: A Study of Chemosensing via the Mouse Grueneberg Ganglion"

_animals, 2022, doi:10.3390/ani12030356_

Round 1

Reviewer 1 Report

The authors present laboratory mouse neurological, physiological and behavioural responses to potential predator derived odours. The methods, results, and conclusions are well and cohesively presented. I cannot comment on the in-vitro methodology but it seems reproduceable.

One minor concern in the behavioural testing is that the authors state stimuli were tested successively. The stimuli should have been presented randomized to help control for associative behaviour 

The study could be improved if the constituent urine chemicals were identified for commonalities and field tested as noted by the authors. Also, understanding the logistical problems involved, open field studies demonstrating avoidance behaviours in wild-type mice would further strengthen the results.    

One additional comment on the methodology -- the authors should provide the sex and ages associated with both the test and stimulus animals. As has been shown in other studies, rodents sex specifically react behaviourally and physiologically to sex hormones in rodent urine. ie adult male mice are attracted to estrogen but avoid testosterone containing urine.

The predator urine likely contains some trace of hormone that might influence the interpretation of results.

Author Response

Response to Reviewer 1

We would like to thank the reviewer for her/his constructive review and very useful comments. We have now addressed all her/his concerns. Please find below our detailed responses.

  • One minor concern in the behavioural testing is that the authors state stimuli were tested successively. The stimuli should have been presented randomized to help control for associative behaviour.

We would like to apologize for the fact that the description of the behavioural test was not clear. Indeed, we avoided the associative behaviours by presenting each mouse with one stimulus/cue per day. Moreover, the non-fear inducing cues (water-rabbit urine) were presented randomly among the fear-inducing cues (mountain lion, raccoon and skunk secretions). We have now changed the Materials and Methods Section accordingly.

  • The study could be improved if the constituent urine chemicals were identified for commonalities and field tested as noted by the authors. Also, understanding the logistical problems involved, open field studies demonstrating avoidance behaviours in wild-type mice would further strengthen the results.

We agree with the reviewer’s comments. A further study of the precise chemical composition of the biological secretions could be indeed performed by chromatography. The identified molecules could be then studied with the methods used in our study, in vitro, ex vivo and in vivo to identify the molecules with kairomone activities 1) on TAS2Rs individually in a heterologous system, 2) on GG neurons and finally 3) on wild-type mice in the lab and open field studies to check their fear-inducing properties. These are indeed long term and very interesting perspectives to our study that go beyond the focus of the present manuscript.

  • One additional comment on the methodology – the authors should provide the sex and ages associated with both the test and stimulus animals. As has been shown in other studies, rodents sex specifically react behaviourally and physiologically to sex hormones in rodent urine. ie adult male mice are attracted to estrogen but avoid testosterone containing urine. The predator urine likely contains some trace of hormone that might influence the interpretation of results.

We agree with the reviewer’s comments and apologize for the missing information about age and sex of both the test and stimulus animals. We have now modified the Materials and Methods Section accordingly.

For the test animals: Adult male mice (C57BL/6) aged between 14 and 19 months were used for behavioural analysis.

For the stimulus animals: We were aware of the possible attractive effects that could be displayed upon estrogen containing biological secretions on male mice. We therefore always mixed multiple samples received from PredatorPee® and Pete Rickard Co. to generate the tested stimuli to make sure they originated from multiple animals. Each mouse was tested with the same set of stimuli.

Reviewer 2 Report

Lopes and coworkers addressed the question whether the three bitter taste receptors TAS2R- 115, 131, 143 present in the Grueneberg ganglion may play a role in the detection of danger-associated olfactory cues. Using in vitro, ex vivo and in vivo experimental approaches, they have found that two new sources of kairomones activate mouse GG neurons. Ca-imaging analyses of transfected HEK293 cells demonstrate that the secretions from skunk and raccoon preferentially activate TAS2R-115 and TAS2R-131, respectively. Ca-imaging analyses on GG slice preparations show that GG neurons recognize secretions from raccoon and skunk, the latter of which produced long lasting responses. Lastly, the authors demonstrate fear-related behavior such as avoidance using these novel stimuli in an open field assay. The study is well structured and the data are convincing.

Only concern:

While the in vitro and ex vivo data directly link to the GG, I have some concerns about the behavioral data. Since the applied stimuli are complex odor mixtures containing many individual compounds it is conceivable that the fear-related behavior observed is a function of the olfactory system (MOE, VNO) and not/not only of the GG. The study would strongly benefit from behavioral experiments on GG axotomized animals. Searching the literature, I see that this technique is well-established in the lab. Alternatively, the authors should elaborate this point in more detail in the discussion.

Author Response

Response to Reviewer 2

We would like to thank the reviewer for her/his constructive review and very useful comments. We have now addressed all her/his concerns. Please find below our detailed responses.

  • While the in vitro and ex vivo data directly link to the GG, I have some concerns about the behavioral data. Since the applied stimuli are complex odor mixtures containing many individual compounds it is conceivable that the fear-related behavior observed is a function of the olfactory system (MOE, VNO) and not/not only of the GG. The study would strongly benefit from behavioral experiments on GG axotomized animals. Searching the literature, I see that this technique is well-established in the lab. Alternatively, the authors should elaborate this point in more detail in the discussion.

We agree with the reviewer’s comment that the behavioural analysis of GG axotomized mice would allow us to selectively implicate the Grueneberg ganglion in the responses observed. As of the present results, we cannot exclude and we are actually convinced of the additional participation of the other olfactory subsystems (MOE and VNO) in the sensing of the biological secretions. The main purpose of this study was to identify two novel sources of kairomones which are fear-inducing molecules and we have previously linked this behaviour to an activated Grueneberg ganglion.  We will now mention these facts in the Discussion section of our manuscript.

As a perspective, it would be interesting to study the chemical composition of the two biological secretions by chromatography to identify specific molecules inducing fear-related behaviours. Once a precise chemical molecule is found, we will perform behavioural analysis on wild-type mice and GG axotomized mice to indeed assess the importance/the selectivity of the Grueneberg ganglion sensing.